# COVID-19—Importance for Patients on the Waiting List and after Kidney Transplantation—A Single Center Evaluation in 2020–2021

**DOI:** 10.3390/pathogens10040429

**Published:** 2021-04-03

**Authors:** Simone C. Boedecker, Pascal Klimpke, Daniel Kraus, Stefan Runkel, Peter R. Galle, Martina Koch, Julia Weinmann-Menke

**Affiliations:** 1Department of Nephrology, University Medical Centre Mainz, Johannes Gutenberg University, 55103 Mainz, Germany; SimoneCosima.Boedecker@unimedizin-mainz.de (S.C.B.); Pascal.Klimpke@unimedizin-mainz.de (P.K.); Daniel.Kraus@unimedizin-mainz.de (D.K.); 2Blood Transfusion Center, University Medical Center Mainz, Johannes-Gutenberg University, 55130 Mainz, Germany; Stefan.Runkel@unimedizin-mainz.de; 3Department of Internal Medicine I, University Medical Centre Mainz, Johannes Gutenberg University, 55130 Mainz, Germany; peter.galle@unimedizin-mainz.de; 4Department of Transplantation Surgery, University Medical Centre Mainz, Johannes Gutenberg University, 55130 Mainz, Germany; martina.Koch@unimedizin-mainz.de

**Keywords:** COVID-19, waiting list, hemodialysis, kidney transplantation

## Abstract

(1) Background: Dialysis patients and recipients of a kidney allograft are at high risk for infection with SARS-CoV-2. It has been shown that the development of potent neutralizing humoral immunity against SARS CoV-2 leads to an increased probability of survival. However, the question of whether immunocompromised patients develop antibodies has not yet been sufficiently investigated; (2) Methods: SARS-CoV-2 antibodies were examined in hemodialysis patients on the waiting list for kidney transplantation as well as patients after kidney transplantation. Patients were interviewed about symptoms and comorbidities, BMI, and smoking history; (3) Results: SARS-CoV-2 antibodies were found in 16 out of 259 patients (6%). The trend of infections here reflects the general course of infection in Germany with a peak in November/December of 2020. Remarkably, patients on the waiting list experienced only mild disease. In contrast, transplanted patients had to be hospitalized but recovered rapidly from COVID-19. Most interesting is that all immunosuppressed patients developed antibodies against SARS-CoV-2 after infection; (4) Conclusions: Even with extensive hygiene concepts, an above-average number of patients were infected with SARS-CoV-2 during the second wave of infections in Germany. Because SARS-CoV-2 infection triggered the formation of antibodies even in these immunocompromised patients, we expect vaccination to be effective in this group of patients. Thus, dialysis patients and patients after kidney transplantation should be given high priority in vaccination programs.

## 1. Introduction

The COVID-19 pandemic has been a global challenge since almost one year. Till end of February 2021, 113,472,187 people have been infected by SARS-CoV-2 with a worldwide mortality of 2.22% [1]. While the first wave of SARS-CoV-2 infections in Germany was comparatively mild with around 190,000 laboratory-confirmed cases by mid-June 2020, the second wave in Q3–4 hit Germany much harder [2]. By end of February 2021, approximately 2.4 million cases had been confirmed and 70,100 deaths associated with COVID 19 reported [3].

The spectrum of clinical manifestations ranges from asymptomatic courses or mild flu-like symptoms to acute respiratory distress syndrome, multiple organ failure, and death [1]. Especially, patients with chronic kidney disease (CKD), dialysis, and organ transplantation are at highest risk of mortality [4]. In this context, the Global Burden of Disease (GBD) called CKD the most common risk factor for severe COVID-19 disease worldwide [5]. Registry data from Europe (European Renal Association—European Dialysis and Transplantation Association) and Canada showed that the mortality rate among dialysis patients with COVID 19 ranges between 20 and 30% [6,7]. This high lethality was also confirmed in our German registry for COVID-19 disease in dialysis patients [8]. There are several reasons for this high mortality. On the one hand, patients with CKD and especially patients on dialysis have a compromised immune system which is due to alterations of the innate and adaptive immunity [9]. On the other hand, these patients are usually older and often suffer from comorbidities such as cardiovascular disease, hypertension, diabetes mellitus, or obesity, which are also known to be associated with increased mortality in COVID-19 disease [4]. Due to this, it is imperative to limit contacts in order to reduce the risk of infection. However, this is not possible in hemodialysis patients due to the need for in-center dialysis treatment three times per week with transportation and long treatment time (4–5 h) in shared spaces with a group of patients as well as the contact to the medical staff, which is unavoidable. In addition, these patients not only have an increased risk of infection, but also represent a significantly increased risk of spreading.

Furthermore, patients after kidney transplantation have a high risk of developing a severe COVID-19 infection because of immunosuppressive therapy and also the presence of comorbidities. Several case series of kidney transplant recipients with COVID-19 have shown an increased mortality [10]. In the retrospective cohort study of the international TANGO consortium, 144 patients hospitalized after kidney transplantation with COVID-19 were analyzed. These patients had pre-existing comorbidities such as hypertension (95%), diabetes (52%), obesity (49%), heart disease (28%), and lung disease (19%). Out of these patients, 52% developed acute graft failure and 29% required mechanical ventilation due to lung failure with an overall mortality rate of 32% [11]. In contrast, initial evaluations of theLEOSS (Lean European open Survey for SARS-CoV-2) registry showed that organ transplantation was not associated with a worse outcome after COVID-19 disease [12].

In connection with severe courses of the disease, the extent of the formation of COVID-19-neutralizing antibodies and the extent to which this is reduced in immunosuppressed collectives are repeatedly discussed. Most recently, Garcia-Beltran et al. have shown that the development of potent neutralizing humoral immunity against SARS-CoV-2 leads to an increased probability of survival and possibly also protects against re-infection [13]. However, the question if immunosuppressed patients form antibodies has not yet been sufficiently investigated.

The aim of our study was to identify the prevalence of SARS-CoV-2 antibodies in our patients on waiting list for kidney transplantation and in patients after kidney transplantation. Furthermore, we compared the outcome of the two collectives after infection and also evaluated dialysis patients who were treated at the University Medical Centre during the pandemic. Finally, we asked whether both first-mentioned immunosuppressed collectives form antibodies after a SARS-CoV-2 infection.

## 2. Results

### 2.1. Patient Characteristics

A total of 259 patients were included in the study: 189 patients on the waiting list, 59 transplanted patients, and 11 dialysis patients not on the waiting list who were treated as inpatients during their COVID-19 disease. During the data collection period, a total of 25 patients were transplanted from the included patients on the waiting list, so these patients appear in both patient collectives. Patient characteristics are summarized in Table 1.

The majority of patients received hemodialysis rather than peritoneal dialysis (Table 2). The mean time on the waiting list was 55 months (Table 2). The transplanted patients had received their allograft on average 42.7 months before inclusion in the study; the mean time on the waiting list before transplantation was 48.9 months (Table 3). This already longer waiting time among patients on the waiting list can be explained by living donations among transplant recipients, some of which were even pre-emptive.

### 2.2. SARS-CoV-2 Infection

SARS-CoV-2 infections occurred in 16 patients on waiting list and kidney transplants groups with a current positive or history of a positive PCR test (see Table 4, Figure 1). Seven waiting list patients reported symptoms related to confirm SARS-CoV-2 infection, as well as eight kidney transplanted patients (see Table 5). One patient on the waiting list had an asymptomatic infection and was isolated at home.

The distribution of infections by quarter reflects the course of infection in Germany. At the beginning of the pandemic, there were only a few cases in the cohorts, with a significant increase in the number of infections during the second wave in November and December 2020 (see in Figure 1).

Of the 16 patients with COVID-19 disease in waiting list and kidney transplant groups, 10 patients had to be treated as inpatients. Seven of these patients were on immunosuppressive therapy due to previous kidney transplantation. We did not see any severe courses of disease leading to an intensive care requirement in patients on the waiting list, but one kidney transplanted patient was admitted to the intensive care unit (ICU). This patient had also received a diagnosis of CNS lymphoma (DLBCL) shortly before his SARS-CoV-2 infection and had already received the first dose of rituximab as part of the chemotherapy. The patient developed multiple organ failure in the course of sepsis with bacterial superinfection and died in a short-term course.

Two other transplanted patients developed acute on chronic graft failure during their COVID-19 disease but could be discharged with an improving kidney function. The other four patients showed mild symptoms like fever or cough and could be discharged after a few days. In three patients, the immunosuppressive therapy was transiently reduced in the context of the COVID-19 disease; therefore, steroids were increased. In one of these patients, m-Tor was briefly paused, in one the CNI level was reduced, and in the ICU patient steroids alone were given, which was also due to the malignant disease. The decision to reduce immunosuppression was made individually by the nephrologists depending on the clinical condition of the patient.

The other patients with SARS-CoV-2 infection on the waiting list usually had very mild courses of disease with only a very low symptom burden. In contrast, most dialysis patients not on the waiting list who had to be hospitalized due to a COVID-19 infection had severe courses of disease. Compared to the transplanted patients, the dialysis patients not on the waiting list were also more severely ill. Four patients died as a result of severe virus pneumonia (Table 5). Overall, the mortality rate among dialysis patients not on the waiting list for kidney transplantation who had to be hospitalized due to their COVID-19 disease was 36%.

### 2.3. SARS-CoV-2 Antibodies

Antibodies were detected in six patients on the waiting list and in seven patients who were seen in our nephrology outpatient clinic or were treated as inpatients (Figure 2). In one case, antibodies were detected in a kidney transplant patient without her being aware of a prior SARS-CoV-2 infection. Remarkably, antibodies were found in all tested patients who had recovered from an infection, including all kidney transplant patients who were all still under triple immunosuppression.

## 3. Discussion

During the period of our investigation, infection with SARS-CoV-2 was detected in 16 patients, of whom 13 also formed autoantibodies. The other three patients could not be tested. Out of all patients, 6% developed SARS-CoV-2 infection which is twice as high as the cumulative incidence of 2.9% in the general population in Germany. This might be caused by the relatively high number of social contacts due to dialysis treatment or visits to the transplant center or due to the close-meshed testing strategies for these patients. Severe courses of COVID-19 disease with intensive care requirements were found in one kidney transplant patient with ongoing immunosuppression and additional rituximab treatment due to lymphoma. No severe courses of disease were observed in patients on the waiting list while patients on dialysis without being on the waiting list for a kidney graft had a COVID-19 mortality of 36%.

The comparison of our data with the previously published data, such as the ERA-EDTA (European Renal Association-European Dialysis and Transplant Association) data, is only possible to a limited extent since the number of infections in the first and second quarters in Germany was significantly lower than in our European neighboring countries or the USA [7,11]. During this period from February till June 2020, in our waiting list cohort, SARS-CoV-2 infection was detected in only one patient out of 189. This leads to the impression that the hygiene concepts worked out by the dialysis centers are efficient in time periods of low numbers of infections. Hygiene concepts at that time included that patients were transported individually to the dialyses, patients were obliged to wear a nose and mouth protection permanently during dialyses, meals were not taken during dialyses, and ventilation concepts were implemented. A weekly SARS-CoV2 antigen testing in the second half of the year supplemented these concepts. Thus, our dialysis patients were not exposed to an increased risk of infection during the first period of the COVID-19 pandemic.

In the second half of the year in which there was a significantly higher increase in the number of infections in Germany, our data also show higher numbers of SARS-CoV-2 infections. In the second half of the year, there was a significant increase in COVID-19 infections during the second wave of the COVID-19 pandemic in Germany. This is also reflected in our analyses with a significant increase in the detection of COVID-19 infections in our cohorts. In relation to the total population, looking at data from the German COVID-19 Registry of Dialysis Patients, dialysis patients are at significantly increased risk for a COVID-19 infection during times of a significantly increased general incidence of infection. The most likely reason for this is that there were reported COVID outbreaks in dialysis centers during this period, with evidence of several positive-tested patients and medical staff. It can be concluded from this that hygiene measures which are sensible and efficient are not sufficient to protect patients from infection in the face of increasing infection rates in a country. Thus, the risk of dialysis patients becoming infected with SARS-CoV-2 raises significantly, making further protective measures necessary for this group of patients, especially by prioritizing vaccination.

The descripted mortality of COVID-19 in patients after kidney transplantation is terrifying and could not be confirmed in our small patient cohort. However, our data are corroborated by published data from the LEOSS Registry, which demonstrated that solid organ transplantation was not a relevant risk factor for a worse outcome after COVID-19 disease [12]. The reason for the significantly better outcome of transplanted patients could be due to the fact that the medical care of these patients was not endangered at any time and patients were able to move into home isolation to reduce potential sources of infection, unlike dialysis patients. In our transplant center, we fortunately treated only eight patients with COVID-19 disease after kidney transplantation and only one patient presented a severe disease and, however, also suffered from an active tumor. However, we are aware that this is definitely substantially below the global average but consistent with data from the LEOSS Registry. A review by Hage et al. also showed that patients who had undergone solid organ transplantation (SOT) did not show an increased risk of infection or mortality. A possible explanation for this was attributed to immunosuppressive therapy which might protect SOT patients from possible overaction of the systemic inflammatory response (cytokine storm), which plays a decisive role in severe COVID-19 disease [14]. This also could support the findings that damping the immune response by using glucocorticoids has proven to have one of the greatest benefits in improving outcome and survival of patients with COVID-19 disease [15]. Another interesting observation is made in the work of Chavarot et al. Here, transplanted patients with COVID-19 disease were matched with other COVID-19 patients who had similar chronic comorbidities and the course of COVID disease was studied. The results showed a similar mortality rate in both collectives, so Chavarot et al. concluded that the severity of COVID-19 disease was not associated with immunosuppression but with the comorbidities [16]. This conclusion confirms results already published in November by Chaudhry et al. [17].

However, it is without doubt that the pandemic is far from being controlled regarding in particular the emerging new mutations and variants of SARS-CoV-2. Therefore, it is essential to develop new therapeutic strategies to protect patients after SOT from severe disease courses [18,19,20,21].

The high mortality rate among transplant recipients in some publications has also repeatedly raised the question whether suspending kidney transplantation is an option due to high infection rates in the relevant region [10]. When comparing the mortality of patients requiring dialysis and those who have already undergone transplantation according to the current state of knowledge, there do not seem to be any major differences with regard to mortality in the two collectives. However, it must be mentioned here that the average age of dialysis patients is significantly higher, and that transplantation is no longer possible in these patients, as well as in patients with severe comorbidities such as, for example, severe heart diseases or tumor diseases. The mortality of COVID-19 disease in patients on the waiting list seems definitely lower than in the entire dialysis patient population. This is confirmed by our data which show that the patients on the waiting list for kidney transplantation mostly had only mild courses of disease and had to be admitted to hospital in significantly fewer cases. In contrast, patients for whom kidney transplantation is no longer a form of therapy showed clearly severe courses of disease with a very high mortality risk.

Thus, it must be questioned whether younger and fitter dialysis patients are exposed to a significantly higher risk of suffering from COVID-19 disease through transplantation. In view of our results, we can clearly answer that the courses of disease were more severe in patients after transplantation than in patients on the waiting list. However, with the exception of one patient who also suffered from a diagnosed cancer, all transplanted patients survived the COVID infection well and none required intensive care treatment. Another argument in favor of transplantation is that the patients will be better able to isolate themselves socially after the transplantation. This could result in a significantly reduced infection rate in this patient group. Thus, overall, the risk of continuing to transplant patients from the waiting list despite the pandemic is acceptable given an adequate supply of organs. Especially in regards to the long waiting time for a kidney transplant in Germany (approx. eight years), a suspension of the transplantation would have severe consequences for the patients on the waiting list. Instead, vaccination against SARS-CoV-2 must be prioritized both for patients who have already been transplanted and for those on the waiting list, as a sufficient vaccination response cannot be expected immediately after transplantation due to immunosuppression.

One note of interest is that our data show that all tested patients on waiting list for kidney transplantation and transplanted patients developed antibodies after a SARS-CoV-2 infection. The detection of antibodies against SARS-CoV-2 in both collectives shows that there is also a detectable immune response in the context of a COVID-19 infection in passively- and actively-immunosuppressed patients. The fact that dialysis patients generally form antibodies after a SARS-CoV-2 infection is also confirmed by recently published data from New York [22]. Thus, this suggests that immunosuppressed patients also develop a vaccination reaction and consequently a certain protection against a severe COVID-19 infection can be assumed. Of interest as well is that one of our transplanted patients had antibodies measured serially over ten months after his SARS-CoV-2 infection disease. Since today, there has been a 60% decrease in the titer over time. Based on this observation, it is certainly useful to determine vaccination titers from patient collectives after vaccination in order to be able to make a statement about the effectiveness and the duration of the vaccination benefit in these special patient collectives.

In summary, vaccination of transplanted patients is recommended and an adequate response to vaccination can be expected. Also with regard to the new SARS-CoV-2 variants, a timely vaccination sesems to be of even greater importance, especially since the first data show that vaccination are also effective here at least in prevention of severe courses of COVID-19 disease [23,24].

## 4. Materials and Methods

In the collective of patients on the waiting list for kidney transplantation, SARS-CoV-2 antibodies were determined from excess material (serum/plasma) during the quarterly antibody screening in all four quarters in 2020. With regard to the patient population after kidney transplantation, SARS-CoV-2 antibodies were examined during the outpatient visit in the period between beginning of January and end of February 2021, as well as the patients we treated as inpatients due to their COVID disease.

A chemiluminescence microparticle immunoassay (CMIA) was used for qualitative detection of IgG and IgM antibodies (Architect SARS-CoV-2-IgG/and SARS-CoV-2-IgM Assays, Abbott GmbH, Wiesbaden, Germany). This assay is specific for antibodies against the nucleocapsid protein of SARS-CoV-2 and does not detect antibodies induced by vaccination. The IgM test, on the other hand, detects antibodies against the spike protein including those induced by vaccination. The specificity of 99.6% and 99.82% for the IgG and the IgM-Assay are indicated by the manufacturer. Furthermore, patients were interviewed by telephone and were questioned with regard to symptoms of SARS-CoV-2 infection, whether hospitalization was necessary, about comorbidities, BMI, and smoking history. In addition to the named collective, the disease courses of dialysis patients who were treated as inpatients in the University Hospital of Mainz due to their COVID disease were evaluated retrospectively.

The primary outcome parameter was the risk of infection for kidney transplant patients and those on the kidney transplant waiting list. Further outcome parameters were the course of disease in these collectives compared with the course of disease in dialysis patients who do not have the possibility of kidney transplantation due to age or comorbidities and the formation of antibodies against SARS-CoV-2 after infection.

## 5. Conclusions

In summary, we conclude that a low incidence of infections in the general population imply that high risk patients are not exposed to an increased risk of infection, which is of great importance especially with regard to dialysis centers. On the other hand, higher infection rates lead to a significant increase in infections in patients who do not have the opportunity for social isolation. Keeping the number of infections low is therefore indispensable for all high-risk populations and should therefore be an urgent political goal.

In addition, we did not find an increased risk of a worse outcome in the renal transplant population after SARS-CoV-2 infection. A possible explanation for this could be the immunosuppressive therapy in these patients which may prevent or suppress the hyperinflammation that occurs during severe COVID-19 infections. Additionally, the significantly higher mortality in some publications could possibly be explained by the overwhelmed capacity of the healthcare system in many regions. Furthermore, the statement that increased mortality in transplanted patients is not related to immunosuppression but to the presence of comorbidities is a very interesting aspect.

Another aspect of our work is that it could be shown that our immunocompromised patient cohorts form antibodies after an infection, and thus there is hope that vaccination might protect these patients, too. In view of the fact that the number of infections in the overall collective is above the German average, prioritized vaccination seems essential in these patients. Especially in the high-risk group of dialysis patients, who due to age and comorbidities do not have the option of transplantation and who cannot isolate themselves through the needed dialysis, vaccination is probably an important option.

However, many questions remain unclear with regard to the COVID-19 pandemic and the vaccines that have only recently become available. However, first answers to these questions can be expected soon. Obviously, risk groups need to be protected more successfully and efficiently, and vaccination seems to be an essential component of this.

## Figures and Tables

**Figure 1 pathogens-10-00429-f001:**
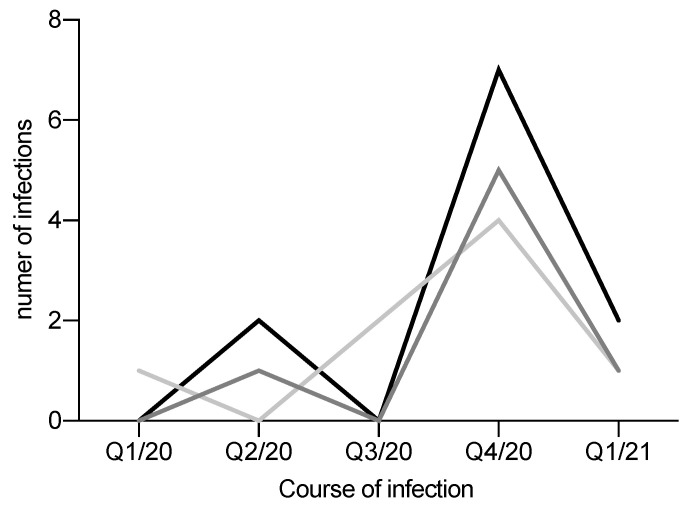
Course of infection (grey: patients on waiting list; light grey: patients after kidney transplantation; black: dialysis patients).

**Figure 2 pathogens-10-00429-f002:**
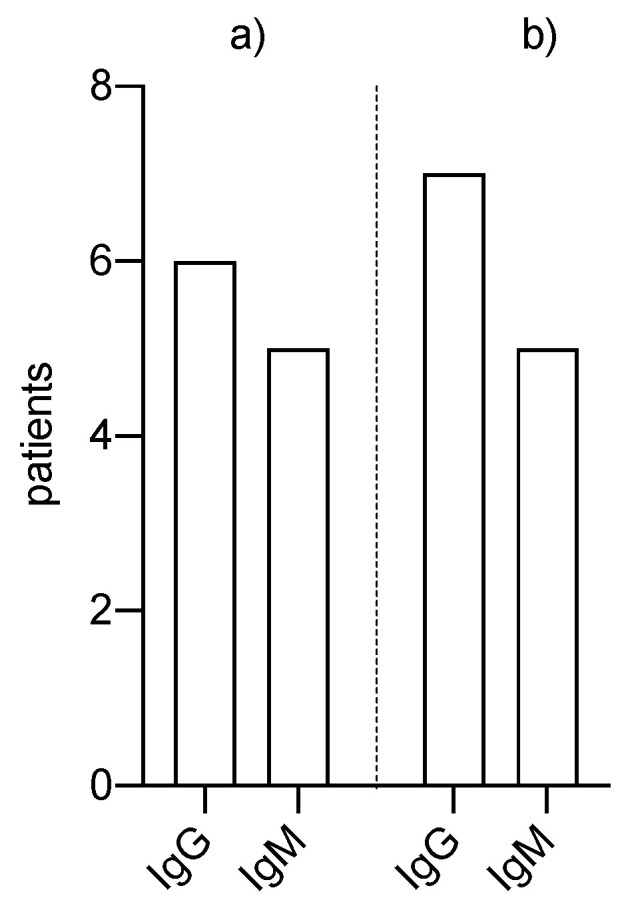
SARS-CoV-2 antibodies in (**a**) patients on the waiting list, and (**b**) kidney transplanted patients.

**Table 1 pathogens-10-00429-t001:** Patient characteristics.

	Patients on Waiting List (*n* = 189)	Kidney Transplants (*n* = 59)	Dialysis Patients with COVID Disease (*n* = 11)
**Age** (in years)	50	53.2	72.7
**Comorbidities** (in patients)			
Overweight (BMI 26–31)	45	22	2
Obesity (BMI > 31)	44	7	4
Diabetes	34	16	5
Hypertension	143	59	9
Heart disease	40	15	6
Chronic Lung disease	12	3	3
Gastrointestinal/Liver disease	11	8	3
Tumor disease	6	2	3
**Smoking**			
PY (pack years)	7.6	5.9	
Active smoker	27	3	
Former smoker	58	12	

**Table 2 pathogens-10-00429-t002:** Dialysis data of patients on waiting list.

	Number
**Dialysis**	
Hemodialysis	165
Peritoneal dialysis	20
Pre-emptive	4
**Time on waiting list** (in month)	55

**Table 3 pathogens-10-00429-t003:** Transplantation data of the kidney transplants.

	Patients Transplanted before 2020	Patients Transplanted in 2020/2021
**Time on waiting list until transplantation** (in month with SD)	42.7 (±35.9)	56.88 (±49.9)
**Time since kidney transplantation** (in month with SD)	68.18 (±76.75)	4.44 (±3.47)
**Immunosuppression**		
Glucocorticoid	32	25
Calcineurin inhibitor	32	24
MMF	17	21
mTOR inhibitor	8	3
other	1	1
**Immunosuppression regime**		
Dual immunosuppression	7	0
Triple immunosuppression	27	25
**SARS-Cov2 infection**	7	1

**Table 4 pathogens-10-00429-t004:** SARS-CoV-2 infection split in quarters in 2021 and in Q1 of 2021 of all patient collectives.

	Patients on Waiting List					Kidney Transplants					Hemodialysis Patients				
	Q1	Q2	Q3	Q4	Q1/21	Q1	Q2	Q3	Q4	Q1/21	Q1	Q2	Q3	Q4	Q1/21
SARS-CoV-2 positive PCR	-	1	-	5	1	1	-	2	5	1	-	2	-	7	2
Asymptomatic infection	-	-	-	1	-	-	-	-	1	-	-	-	-		-
Hospitalization	-	-	-	3	-	1	-	1	4	1	-	2	-	7	2
Treatment on ICU	-	-	-	-	-	-	-	-	-	1	-	1	-	1	-
Mechanical ventilation	-	-	-	-	-	-	-	-	-	1	-	-	-	1	-
Death	-	-	-	-	-	-	-	-	-	1	-	-	-	3	1

**Table 5 pathogens-10-00429-t005:** COVID-19 characteristics.

	Patients on Waiting List	Kidney Transplants	Dialysis Patients
Congestion of the nose	1	1	-
Headache	1		-
Cough	3	6	8
Sputum	-	1	5
Shortness of breath	1	1	6
Pneumonia	1	1	6
Fever	3	6	7
Gastrointestinal disorders	-	1	-
Muscle pain	-		1
Acute kidney damage	-	3	-
Multiorgan failure	-	1	1

## Data Availability

Not applicable.

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
