# Peer review of "COVID-19—Importance for Patients on the Waiting List and after Kidney Transplantation—A Single Center Evaluation in 2020–2021"

_pathogens, 2021, doi:10.3390/pathogens10040429_

Round 1
Reviewer 1 Report
During world-wide COVID-19 pandemic the article “COVID-19 – Importance for patients on the waiting list and after kidney transplantation – a single center evaluation in 2020-2021” is of high importance. The article suggest kidney recipients, patients on waiting list for kidney transplantation and patients undergoing dialysis to give priority for vaccination as immunocompromised patients. However, I have some notes:
Scope. In this special issue “Pathogens” wants to encourage authors to submit work elucidating the role of B- and T-cells, dysfunctional T- and B-cell regulation in systemic autoimmune diseases and immunosuppressed patients with special regard to opportunistic infections. I think the article is a little bit out of this scope. There is no elucidation of the role of B- and T-cells, dysfunction and etc.. However, the article demonstrates the epidemiological situation in Germany on immunocompromised patients under the hot topic COVID-19 infection, and available prevention by vaccination and its role.
Abstract can be improved. I suggest to add these sentences: „Garcia-Beltran et al. have shown that the development of potent neutralizing humoral immunity against SARS CoV-2 leads to an increased probability of survival and possibly also protects against reinfection. However, the question if immunosuppressed patients form antibodies has not yet been sufficiently investigated.” Or the abbreviated main aim.
Results. It is written that only 16 patients had positive PCR tests. Table 4 is not clear, legend with an explanation might help to understand the data better. Moreover, not all 16 patients are discussed, one from kidney transplant patient group is missing, why?
There is no information about the antibodies’ levels, was there any relation with a severity of COVID-19 infection? Was there any relation with a medicament they receive?
Patients were interviewed, but any correlation analysis positive/negative are given. Example: does smoking affect antibodies level? Or similar...
Author Response
Thank you for this constructive review. We have addressed each of the concerns in this response and within the revised manuscript. The comments by the reviewer are in bold type followed by the response in regular type. Moreover, the changes within the manuscript are in red. You can find the corrected manuscript in the attachment.
1. Scope. In this special issue “Pathogens” wants to encourage authors to submit work elucidating the role of B- and T-cells, dysfunctional T- and B-cell regulation in systemic autoimmune diseases and immunosuppressed patients with special regard to opportunistic infections. I think the article is a little bit out of this scope. There is no elucidation of the role of B- and T-cells, dysfunction and etc.. However, the article demonstrates the epidemiological situation in Germany on immunocompromised patients under the hot topic COVID-19 infection, and available prevention by vaccination and its role.
Thank you very much for your supporting comment that this topic is very important for immunocompromised patients.
2. Abstract can be improved. I suggest to add these sentences: „Garcia-Beltran et al. have shown that the development of potent neutralizing humoral immunity against SARS CoV-2 leads to an increased probability of survival and possibly also protects against reinfection. However, the question if immunosuppressed patients form antibodies has not yet been sufficiently investigated.” Or the abbreviated main aim.
Thank you for the helpful comment. The abstract has been rewritten as you suggested. (Line 18 – 42)
3. It is written that only 16 patients had positive PCR tests. Table 4 is not clear, legend with an explanation might help to understand the data better. Moreover, not all 16 patients are discussed, one from kidney transplant patient group is missing, why?
We must apologies for incorrectly listing 15 patients, we corrected it to 16 patients in the manuscript, see line 135. Also, the legend of Table 4 has been adjusted so that it is clear that all patients with SARS-CoV2 infection are mentioned in this table and not only the 16 positive patients on the waiting list and the transplanted patients. (Line 137)
4. There is no information about the antibodies’ levels, was there any relation with a severity of COVID-19 infection? Was there any relation with a medicament they receive?
This would indeed be a very interesting point, but unfortunately, due to the test used (see material and methodology), we do not carry out titer determinations in our laboratory, but only determine the index at which a value > 1.4 is considered positive.
5. Patients were interviewed, but any correlation analysis positive/negative are given. Example: does smoking affect antibodies level? Or similar...
Thank you for this comment. As already explained, these analyses are unfortunately not possible at the present time, as the levels of the antibodies are not available.

Reviewer 2 Report
Great article about a very relevant and interesting topic

Author Response
Thank you for this comprehensive and constructive review. We have addressed each of the concerns in this response and within the revised manuscript. The comments by the reviewer are in bold type followed by the response in regular type. Moreover, the changes within the manuscript are in red. The corrected manuscript can be find in the attachment.
1. Lines 102-103: i would recommend including the 25 patients who were transplanted from the waiting list in one but not both groups (eg, would place on the “kidney transplant” list and not on the “patients on waiting list”). Including them in both groups skews the results you obtained
Thank you for the comment. We understand your concerns and have thought about this point for a long time. However, in the end we decided to keep the 2020 transplanted patients in both collectives, as all patients had several antibody determinations during their waiting period for kidney transplantation and this data would be lost if we only listed these patients in the collective of transplanted patients. The consequence would then be that, in our opinion, the data would have been incompletely evaluated.
2. Table 1: please clarify what “PY” stands for
Thank you for this comment. We implemented this recommendation in the manuscript.
3. Lines 108-111 and Table 2: please clarify what the “mean time on the waiting list was 55 months” is as compared to “waiting list before transplantation of 48.9 months.”
Following your comment, we implement an explanation, why patients on the waiting list currently have a longer waiting time for transplantation than patients who have already been transplanted, see lines 113-115.
4. Does table 2 include the 25 patients who were transplanted in the middle of the data collection period as well? If yes, then the numbers reported for data in table 3 are likely very skewed from including these 25 patients. It may be best to separate the 25 newly transplanted patients from the prior 34 patients in the transplant cohort. It would be interesting to do so so that you may see if there are any differences between the “fresh transplants” and the “older transplants.”
Thank you for the comment. Indeed, the 2020 transplanted patients also appear here in both tables. Based on this, we have accepted your suggestion and implemented this interesting point. You can now find the new data in Table 3 in the manuscript.
5. Line 133: 15 patients from which group? Please clarify
Thank you for this helpful comment, we add the groups to the sentence and we must apologize for incorrectly listing 15 patients instead of the 16 patients mentioned above.
6.Line 141-147: Was there a protocol used for immunosuppression regimen chosen or was it nephrologist specific decision?
In case of reduction of immunosuppression, individual decisions were made by nephrologists depending on the patient's condition. Here, however, recommendations were followed based on previous experience to stop antiproliferative agents like MMF or Azathioprin first. However, if severe COVID-19 disease a reduction or even a withdrawal of the CNI is also appropriate. This information was added to the manuscript. (Line 152-154)
7. Lines 152-153: not listed in table 5
Thank you very much for this comment. We have added pneumonia to table 5.
8. Lines 185-189: are the waiting list patients all on dialysis? please clarify this earlier in the paper. Also, it would be helpful to know what the “hygiene concepts” are that worked.
Thank you for the point about the hygiene concepts. We implemented this recommendation in the text, see lines 194 - 198. In Table 2 you can find a more detailed differentiation of which patients on the waiting list are doing peritoneal dialysis or hemodialysis, also listed here is that 4 patients on the waiting list are currently listed as pre-emptive, so that no dialysis is currently being performed in these patients.
9. Line 211: that’s impressive! Please describe your workflow or how you protected medical care for transplanted patients as this is not a common experience post-COVID. Many patients had disrupted medical care so it would be interesting to learn your process.
Due to the fact that the health care system in Germany has not reached its limits so far in the SARS-COV2 pandemic, post-transplant care has been fully continued and not omitted. During the outpatient appointments, strict hygiene measures were observed and the number of patients per day was reduced so that there were no long waiting times. For patients who had concerns about an outpatients visit in the University hospital, primary care physicians were involved in blood collection, and telephone rounds were conducted when laboratory results were available.
10. Line 252: please clarify sentence: more severe how?
As recommended, we add examples in the manuscript to severe comorbidities for a better understanding. You can find it in line 254-355.
11. Line 333: would be careful about such strong wording as “mandatory” as it raises many ethical and legal questions about vaccination mandates and patients’ choices/preferences, especially in light of the fact that we don’t have enough data to show that the benefit of vaccination far outweighs the risks (yet).
Thank you very much for this comment. Based on this, the sentence was changed and rewritten as “mandatory” to “possibly an important option”. (Line 345)
